

# Reconstructing the demographic history of divergence between European river and brook lampreys using approximate Bayesian computations

Quentin Rougemont[1,2], Camille Roux[3], Samuel Neuenschwander[3,4], Jérôme Goudet[3], Sophie Launey[1,2] and Guillaume Evanno[1,2]

[1] UMR 985 Ecologie et Santé des Ecosystèmes, Institut National de la Recherche Agronomique, Rennes, France
[2] UMR 985 Ecologie et Santé des Ecosystèmes, Agrocampus Ouest, Rennes, France
[3] Department of Ecology and Evolution, Université de Lausanne, Lausanne, Switzerland
[4] Swiss Institute of Bioinformatics, Vital-IT, Lausanne, Switzerland

## ABSTRACT

Inferring the history of isolation and gene flow during species divergence is a central question in evolutionary biology. The European river lamprey (*Lampetra fluviatilis*) and brook lamprey *(L. planeri)* show a low reproductive isolation but have highly distinct life histories, the former being parasitic-anadromous and the latter non-parasitic and freshwater resident. Here we used microsatellite data from six replicated population pairs to reconstruct their history of divergence using an approximate Bayesian computation framework combined with a random forest model. In most population pairs, scenarios of divergence with recent isolation were outcompeted by scenarios proposing ongoing gene flow, namely the Secondary Contact (SC) and Isolation with Migration (IM) models. The estimation of demographic parameters under the SC model indicated a time of secondary contact close to the time of speciation, explaining why SC and IM models could not be discriminated. In case of an ancient secondary contact, the historical signal of divergence is lost and neutral markers converge to the same equilibrium as under the less parameterized model allowing ongoing gene flow. Our results imply that models of secondary contacts should be systematically compared to models of divergence with gene flow; given the difficulty to discriminate among these models, we suggest that genome-wide data are needed to adequately reconstruct divergence history.

## INTRODUCTION

Understanding the spatio-temporal conditions favouring species emergence is a fundamental question in evolutionary biology. One long standing controversy concerns the geographical setting promoting species divergence (*Butlin, Galindo & Grahame, 2008*; *Fitzpatrick, Fordyce & Gavrilets, 2008*). Theory predicts that the accumulation of genetic incompatibilities is rather straightforward under allopatric conditions without

Corresponding author
Quentin Rougemont, quentinrougemont@orange.fr

gene-flow (*Turelli, Barton & Coyne, 2001*; *Coyne & Orr, 2004*; *Barton & de Cara, 2009*). In contrast, speciation with gene flow theoretically requires (i) strong divergent selection and non-random mating, (ii) high genetic variance and (iii) non-random association of traits under disruptive selection and those involved in assortative mating (*Dieckmann & Doebeli, 1999*; *Gavrilets, 2003*; *Gavrilets, 2014*; *Coyne & Orr, 2004*). Importantly, the current geographical distribution of contemporary species may not reflect the initial conditions of divergence as most species may have undergone alternative phases of splits and contacts over historical periods (*Hewitt, 1996*; *Hewitt, 2004*; *Hewitt, 2011*; *Bierne et al., 2011*). As a result, reconstructing the history of demographic events that have shaped the genetic architecture of present-day populations is of primary importance to understand how speciation operates and infer the role of gene flow during divergence. The accuracy of this reconstruction will depend on an adequate statistical method for demographic inferences, but also on the relevance of the sampling scheme.

Simulation-based methods are helpful for inferences although the tested models often do not entirely reflect the complexity of the usually unknown demographic history of the populations studied (*Wakeley, 2008*). For instance, several studies using full likelihood approaches implemented in the IM and IMa programs (*Hey & Nielsen, 2004*; *Hey & Nielsen, 2007*; *Hey, 2010*) have compared isolation with migration (IM) models against a model of strict isolation (SI) and revealed a widespread effect of gene flow during divergence (e.g., *Pinho, Harris & Ferrand, 2008*; *Niemiller, Fitzpatrick & Miller, 2008*; *Strasburg & Rieseberg, 2008*). However, this method makes a number of simplifying assumptions (*Strasburg & Rieseberg, 2010*; *Strasburg & Rieseberg, 2011*) and does not allow reconstructing complex scenarios with several parameters, due to computation burden or intractable likelihood computation. Thus the complexity of demographic events may have been missed (*Nielsen & Wakeley, 2001*; *Hey, 2010*). For example, most of these studies failed to distinguish between primary *versus* secondary differentiation (i.e., allopatric divergence followed by secondary contacts) hence no general conclusion about the ubiquity of either mechanism during speciation could be drawn yet. Recent advances in coalescent theory (*Wakeley, 2008*) and Bayesian methods (*Tavare et al., 1997*; *Beaumont, Zhang & Balding, 2002*; *Beaumont, 2010*; *Csilléry et al., 2010*) now allow for explicit tests of alternative and complex models of divergence. In particular, approximate Bayesian computation (ABC) bypasses the need to compute full likelihoods, as this is not possible or computationally too intensive for complex models with many parameters and large datasets (*Beaumont, Zhang & Balding, 2002*). ABC has been used with success to test alternative models of divergence in various taxa and provided useful information on the level of interspecific introgression and complexity of demographic history underlying population divergence (*Fagundes et al., 2007*; *Duvaux et al., 2011*; *Roux et al., 2013*; *Roux et al., 2014*; *Nadachowska-Brzyska et al., 2013*; *Nater et al., 2015*).

Several studies have focussed on a single population pair to infer the history of divergence but drawing general conclusions may be complicated with such a reduced sampling scheme. On the other hand, studies of replicated pairs of diverging populations have proven very useful to understand the genetic mechanisms of divergence and speciation by showing that populations can independently evolve similar reproductive barriers in the face of ongoing

gene-flow (e.g., *Schluter & McPhail, 1993*; *Nosil, Crespi & Sandoval, 2002*; *Colosimo et al., 2005*; *Johannesson et al., 2010*). Such results were generally interpreted as evidence for parallel adaptation of diverging populations due to the recent action of natural selection. However, alternative scenarios of divergence including secondary contacts after periods of allopatry have rarely been investigated (*Bierne, Gagnaire & David, 2013*; *Butlin et al., 2014*; *Welch & Jiggins, 2014*).

Lampreys are jawless vertebrates (agnathans) thought to have diverged from the gnathostomes lineage (jawed vertebrates) approximately 590 million years ago (*Hedges et al., 2015*). Most lampreys occur as pairs of closely related species reproducing in the same rivers but with very distinct adult life history strategies. Within such pairs, one taxon migrates at sea and adopts a parasitic-hematophagous life style while the other is freshwater-resident, non-parasitic and does not feed at the adult stage (*Docker, 2009*). Despite a large number of evolutionary and developmental studies in lampreys (*Heimberg et al., 2010*; *Shimeld & Donoghue, 2012*; *Smith et al., 2013*; *Lagadec et al., 2015*), there is a high uncertainty about taxonomic relationships among these so called lamprey 'paired' species (*Docker, 2009*). For instance, the European river lamprey (*Lampetra fluviatilis*) and brook lamprey (*L. planeri*) display marked morphological differences at the adult stage: individuals of the anadromous and parasitic river lamprey are on average 2.2 times longer than resident and non-parasitic brook lampreys but adults of both species can be found on the same spawning ground (*Lasne, Sabatié & Evanno, 2010*) and this size difference likely forms the most important prezygotic barrier to gene-flow (*Beamish & Neville, 1992*; *Rougemont et al., 2015*). The genetic differentiation between these two taxa is usually low when measured either with allozymes (*Schreiber & Engelhorn, 1998*), mtDNA (*Espanhol, Almeida & Alves, 2007*; *Blank, Jürss & Bastrop, 2008*) or microsatellites markers (*Bracken et al., 2015*; *Rougemont et al., 2015*) and these species have been hypothesized to be different ecotypes of a single species (*Docker, 2009*). Only two studies, based on a single population pair, reported a strong differentiation between *L. planeri* and *L. fluviatilis* (*Mateus et al., 2013*; *Mateus et al., 2016*). The only large scale phylogeographic study based on mtDNA revealed a very low level of divergence that was hypothesized to result from ongoing gene flow or very recent divergence following postglacial dispersion (*Espanhol, Almeida & Alves, 2007*; *Mateus et al., 2016*). However, it is known that widespread mtDNA introgression among sympatric taxa can easily obscure their taxonomic relationships (*Shaw, 2002*). However, phylogeographic approaches do not allow contrasting alternative scenarios of divergence and do not address gene flow following divergence. As a consequence, relatively little is known so far on the history of divergence between *L. fluviatilis* and *L. planeri* and most studies concluded on a prominent role of recent postglacial divergence (*Espanhol, Almeida & Alves, 2007*; *Bracken et al., 2015*) or ecological processes (*Salewski, 2003*). Overall, few studies have used a wide number of pairs of river and brook lamprey connected by gene flow and realistic scenarios of demographic history have never been modelled.

Recently *Rougemont et al. (2015)* studied ten pairs of sympatric and parapatric populations of *L. fluviatilis* and *L. planeri* and found varying levels of genetic differentiation ranging from strong gene flow ($F_{ST} = 0.008$) to important genetic differentiation

($F_{ST} = 0.189$) depending on population pairs. They concluded that these two species may represent partially reproductively isolated ecotypes, a statement that was consistent with the low degree of reproductive isolation measured in experimental crosses (*Hume et al., 2013*; *Rougemont et al., 2015*). However, this pattern of low genetic differentiation observed in several population pairs can be explained by two opposite hypotheses: (i) ongoing divergence with gene flow or (ii) secondary contact after a period of allopatry that did not allow the accumulation of sufficient reproductive barriers, including endogenic barriers. Here we investigated the demographic history of divergence between river and brook lampreys by testing five models of divergence including these two competing scenarios with an ABC approach based on microsatellite data from six population pairs of *Lampetra*.

## MATERIALS AND METHODS

### Sampling and genotyping

*L. fluviatilis* and *L. planeri* samples were collected from 2010 to 2014 in 6 population pairs from northern France (data from *Rougemont et al., 2015* and Fig. 1). Three pairs coexist in sympatry (Aa, Bethune and Oir Rivers) and 2 pairs are not strictly sympatric because a small obstacle occurs on the first river (Hem), while in the second case, populations are located 8 km apart, on the same stream section (Bresle). The last pair is parapatric (Risle), but with a moderate $F_{ST}$ value similar to what is observed in sympatric populations (*Rougemont et al., 2015*). We focused on pairs highly connected by gene flow previously identified in Rougemont et al., study because in disconnected pairs, inferences on the long term history of divergence may be biased by the recent effect of genetic drift in isolated *L. planeri* populations. In parapatric situations *L. planeri* populations were generally geographically isolated in upper parts of rivers due to anthropogenic barriers to migration with no opportunity for gene flow with *L. fluviatilis*. The sampling included temporal replicates on the Oir (2010, 2011 and 2014), Bresle (2011 and 2014), and Risle rivers (2011 and 2014). A set of 13 microsatellites was used to genotype a total of 727 individuals following the protocol described in *Gaigher et al. (2013)*.

### Summary statistics

Given the lack of genetic differentiation between samples collected in different years in the same river, they were merged together (*Rougemont et al., 2015*). Similarly, we pooled brook lamprey individuals sampled in upstream and downstream areas on the Aa and Hem river as these two groups were not significantly differentiated (Table S1). We also pooled river lamprey individuals from the Aa and Hem rivers given the lack of differentiation between these populations (*Rougemont et al., 2015*). Summary statistics were then computed for each pooled sample. As summary statistics used for comparison between simulated and observed datasets, we computed the average and standard deviation values of: the number of alleles (A), Allelic richness (Ar), observed and expected heterozygosity (Ho and He), allele size in base pairs, the Garza-Williamson index (GW, *Garza & Williamson, 2001*), $G_{ST}$ (Nei1, 973) and $\delta\mu^2$ (*Goldstein et al., 1995*). The GW index (M ratio) aims at assessing reductions in effective population size and is calculated as $M = \frac{A}{R+1}$ with
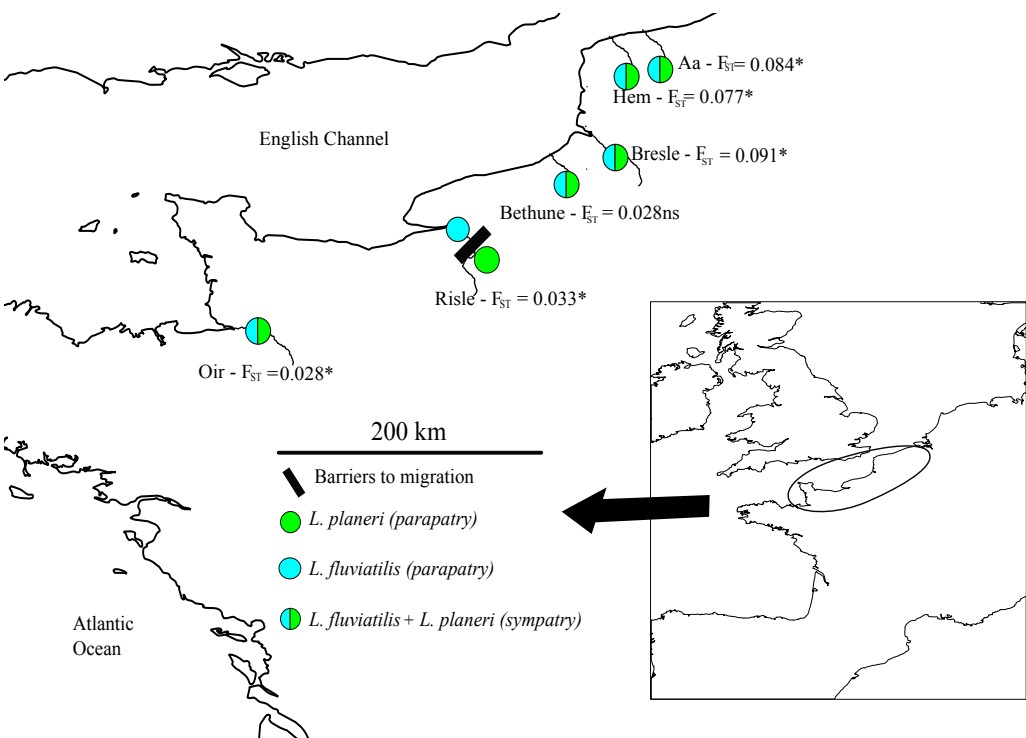

**Figure 1** **Map of sampling sites across the channel area.** River names match those given in Table 2 and $F_{st}$ values are given for each population pair.

$A$ is the number of alleles for a given loci in each population and R is the allelic range. Following (*Excoffier, Estoup & Cornuet, 2005*) we add 1 to the denominator to avoided dividing by zero in case of a monomorphic sample. Goldstein et al. $\delta\mu^2$ is a measure of genetic distance developed for microsatellite data defined as $\delta\mu2 = (\mu_1 - \mu_2)^2$ where $\mu_1$ and $\mu_2$ represents the average number of allelic size differences within populations 1 and populations 2 respectively. Each statistics was computed within populations as well as globally except for the Gst and $\delta\mu^2$ which are pairwise statistics. All statistics were computed using R scripts (*R Development Core Team, 2011*) available on github at https://github.com/QuentinRougemont/MicrosatDemogInference.

## Testing alternatives demographic scenario
### ABC coalescent simulations

For each population pair we used an approximate Bayesian computation (*Beaumont, Zhang & Balding, 2002*; *Csilléry et al., 2010*) framework to statistically compare five alternative models of demographic history (Fig. 2): (1) the two studied populations derive from a single panmictic gene pool (PAN); (2) a strict isolation model between sister populations (SI); (3) an isolation with migration model (IM); (4) a model allowing ancient migration but recent isolation (AM) and (5) a model of secondary contact after past isolation (SC). The PAN model assumes a single panmictic population with constant population size. The SI model assumes a strict and instantaneous split of the ancestral population into two daughter populations with constant size and no subsequent gene-flow.

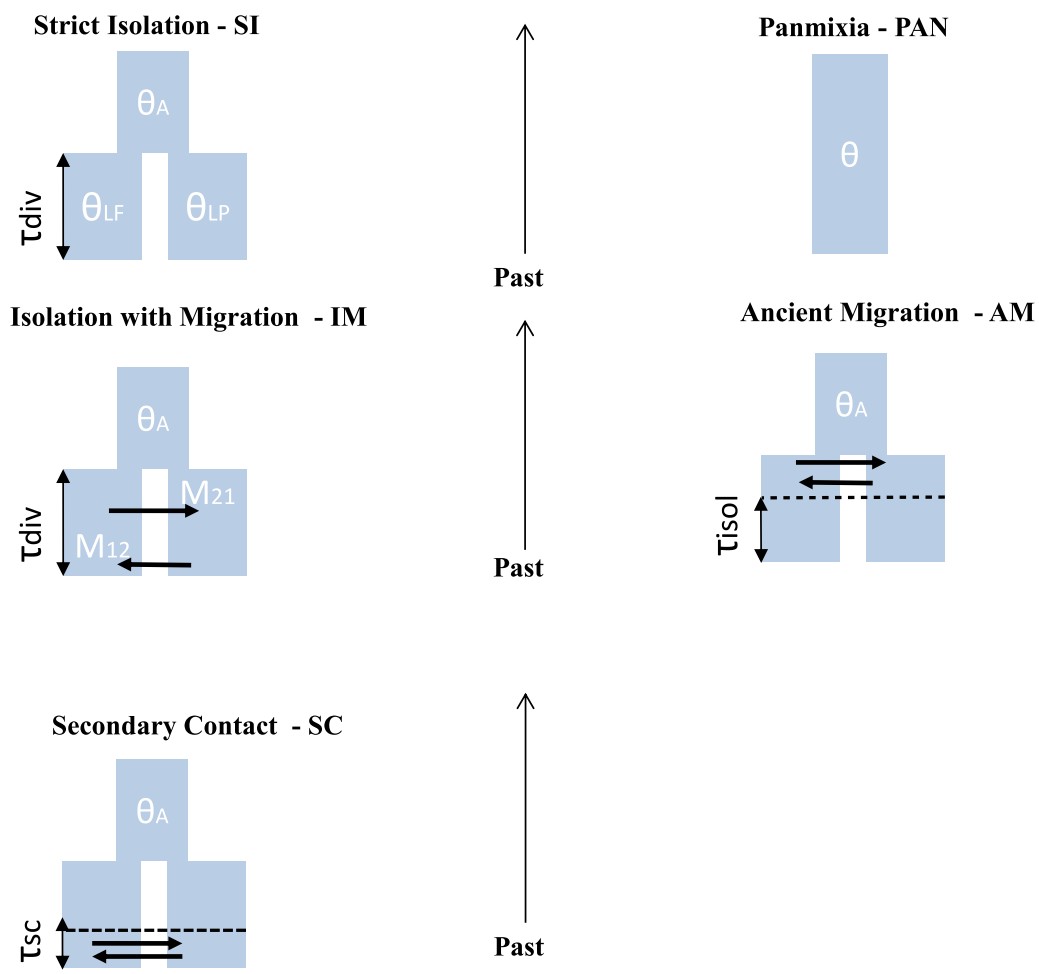

**Figure 2  Different scenario of divergence between *L. planeri* and *L. fluviatilis*.** Five models with different parameters are tested and compared. Two null models: stict Isolation (SI) and Panmixia (PAN). Three models of migration: isolation with constant migration (IM), ancient migration (AM) and secondary contact (SC). The following parameters are shared by all models: $\tau_{\mathrm{div}}$ : number of generations since divergence time. $\theta_A, \theta_{Lf}, \theta_{Lp}$: effective population size of the ancestral population, of *L. fluviatilis* and *L. planeri* respectively. $\tau_{\mathrm{isol}}$ is the number of generations since the two ecotypes have stopped exchanging genes. $\tau_{sc}$ is the number of generations since the two ecotypes have entered into a secondary contact after a period of isolation. $M_{12}$ and $M_{21}$ represent the number of migrants expressed in $4.Nm$ units per generation with $m$ the proportion of population made of migrants from the other populations.

The IM model assumes continuous gene-flow between daughter populations after the initial split at constant rate over generations. The AM model assumes gene-flow between the two diverging populations during the first generations following the split of the ancestral population. The SC model describes the split of an ancestral population in two isolated daughter populations, the two evolving lineages then experience gene flow through a secondary contact starting $T_{SC}$ generations ago. For IM, AM and SC models, the number of migrants was scaled by $M = 4.N_0.m$, with $M_1$ the number of migrants from *L. fluviatilis* to *L. planeri* and $M_2$ the number of migrants from *L. planeri* to *L. fluviatilis* and $m$ the fraction of the population made of migrants from the other population at each generation.

Coalescent simulations were performed using the ms software (*Hudson, 2002*) assuming an infinite-site model of mutation, in which most parameters are scaled by the effective population size of an arbitrarily chosen reference population ($N_{\text{ref}}$) with impact on conclusions drawn by the ABC analysis. Each model was also characterized by a scaled effective population size $\theta : \theta_A/\theta_{\text{Ref}}, \theta_{lf}/\theta_{\text{Ref}}, \theta_{lp}/\theta_{\text{Ref}}$ where $\theta_{\text{Ref}} = 4N_{\text{Ref}}\mu, \mu$ representing the mutation rate per generation. Patterns of genetic diversity suggested that river lamprey populations display a greater *Ne* than populations of brook lamprey (*Rougemont et al., 2015*). Thus, $\theta_{lf}$ was sampled on the interval 0–3 and $\theta_{lp}$ in the interval 0-max ($\theta_{lf}$). $\theta_{\text{Ref}}$ was set to 1 (i.e., we assumed $N_{\text{Ref}} = 1,000$ and $\mu = 2.5e^{-4}$). $N_{\text{ref}}$ was chosen based on prior knowledge of possible population size in lampreys (Q Rougemont et al., 2015, unpublished data). $\mu$ was chosen according to values frequently observed in fishes (*Shimoda et al., 1999*; *Steinberg et al., 2002*; *Yue, David & Orban, 2006*) and other vertebrates species (e.g., *Nance et al., 2011*). The panmictic model was only characterized by the unique effective mutation rate $\theta$ which was also modelled on the interval 0–3. All models (except PAN) also incorporated the scaled time of divergence, $\tau_{\text{split}}/4N_{\text{Ref}}$, where $\tau_{\text{split}}$ is the time measured in number of generations and drawn from a uniform distribution in the interval 0–25. The two parameters $\tau_{\text{iso}}$ (AM model) and $\tau_{sc}$ (SC model) were computed from uniform distributions defined on the interval 0–$\tau_{\text{split}}$. Since the genetics and ecology of lampreys is poorly known, we chose to include commonly used parameters from the literature (*Pinho & Hey, 2010*) after exploring different combinations of uninformative priors following *Cornuet, Ravigné & Estoup (2010)* (Table 1). Binary simulated data from ms were converted into microsatellite data using a generalised stepwise mutation model (GSM) in which the probability of changes of the repeat number in each mutation event was modelled by a geometrical parameter $\alpha$ distributed following a uniform prior distribution sampled on the interval 0–0.5. All computations were run in R and took into account differences in sample size for each of the thirteen loci. Summary statistics were computed from the transformed microsatellite data. One million simulations composed of the thirteen microsatellite loci were computed under each demographic model. All R code used for ABC computation is available at https://github.com/QuentinRougemont/MicrosatDemogInference.

## Model selection
### ABC approach
We evaluated the posterior probabilities of each demographic model using an ABC framework implemented in the abc package in R (*Csilléry, François & Blum, 2012*). We compared all models at a time by computing posterior probabilities using a feed forward neural network based on a nonlinear conditional heteroscedastic regression in which the model is considered as an additional parameter to be inferred. This procedure allows taking into account correlations of summary statistics and distortion hence reducing the problem of curse of dimensionality (*Blum & François, 2010*). In the rejection step, we retained the 0.02% simulations closest to the observed summary statistics, which were subsequently weighted by an Epanechnikov kernel that peaks when $S_{\text{obs}} = S_{\text{sim}}$. The regression step was performed using 50 neural networks and 15 hidden layers.

**Table 1** **Prior for all models.** $\theta_A, \theta_1, \theta_2$ = effective mutation rate for the ancestral, river lamprey and brook lamprey populations respectively. $M_1, M_2, M_{Anc}$ = *Effective migration rate for the ancestral, river lamprey and brook lamprey populations respectively. $\tau$ = divergence time, $\tau_{isol} \tau_{sc}$ divergence time under the ancient migration model and time of secondary contact respectively.*

| Parameters | Models | Prior |
|---|---|---|
| $\theta_A = 4N_{Anc}\mu$ | SI, IM, AM, SC | Uniform [0–3] |
| $\theta_1 = 4N_1\mu$ | SI, IM, AM, SC, PAN | Uniform [0–3] |
| $\theta_2 = 4N_2\mu$ | SI, IM, AM, SC | Uniform [0–($\theta_1$)] |
| $M_1 = M_2 = 4N_1m$ | IM, SC | Uniform [0–20] |
| $M_{ANC} = 4N_1m$ | AM | Uniform [0–20] |
| $\tau = 4N_1t$ | SI, IM, AM, SC | Uniform [0–25] |
| $\tau_{isol} = 4N_1t$ | AM | Uniform [0–$\tau$] |
| $\tau_{sc} = 4N_1t$ | SC | Uniform [0–$\tau$] |

Notes.
SI, strict isolation; IM, isolation with migration; AM, ancient migration; PAN, Panmixia; SC, secondary contact model.

**Table 2** **Estimates of populations genetic parameters for each pair of river and brook lamprey populations.** $N$, number of individuals used for ABC analysis; Ar, Allelic richness; He, expected heterozygosity; GW, Garza-Williamson Index. Population are classified by increasing order of genetic differentiation.

| Pop | River name | $N_{Lf}$ | $N_{Lp}$ | $F_{ST}$ | Ar $_{Lf}$ | Ar $_{Lp}$ | He $_{Lf}$ | He $_{Lp}$ | GW $_{Lf}$ | GW $_{Lp}$ | $\Delta\mu^2$ |
|---|---|---|---|---|---|---|---|---|---|---|---|
| OIR | Oir | 104 | 74 | 0.028 | 4.45 | 3.61 | 0.52 | 0.508 | 0.525 | 0.622 | 0.204 |
| BET | Bethune | 14 | 14 | 0.028 | 3.51 | 3.36 | 0.516 | 0.471 | 0.452 | 0.464 | 0.507 |
| RIS | Risle | 75 | 75 | 0.033 | 3.84 | 3.92 | 0.503 | 0.472 | 0.497 | 0.421 | 0.842 |
| HEM | Hem | 30[a] | 65[b] | 0.077 | 4.21 | 3.53 | 0.504 | 0.477 | 0.406 | 0.487 | 1.633 |
| AA | Aa | 34[a] | 69[b] | 0.084 | 4.21 | 3.76 | 0.514 | 0.522 | 0.406 | 0.505 | 0.915 |
| BRE | Bresle | 93 | 80 | 0.091 | 4.14 | 4.91 | 0.49 | 0.49 | 0.466 | 0.263 | 34.37 |

Notes.
[a]For the ABC inference, individuals of river lamprey from the AA and Hem ($F_{ST} = 0$) river were pooled together to obtain a sample size similar to the one of brook lampreys.
[b]Brook lamprey samples from the AA and Hem rivers are composed of upstream and downstream samples from the *Rougemont et al. (2015)* study.

## ABC cross-validation

We performed model checking to compute the classification error rate of the inferred model using pseudo-observed simulated datasets (PODS). We randomly selected 1,000 PODS from one million of simulations computed under each simulated model. We used the same ABC selection procedure as above to estimate the probability of the PODS. Knowing the true model we then computed the type I error rate that corresponds to the risk of excluding the previously inferred scenario when it is the true and the type II error rate that corresponds to the risk of selecting the previously inferred scenario when it is false following *Cornuet, Ravigné & Estoup (2010)*. The most important here was the type II error that, with regards to the selected dataset, corresponds to the risks of erroneously selecting the focal scenario. We validated the accuracy of our procedure by performing the same analysis in pairwise model comparisons of all models of divergence with gene flow (AM, IM, SC) against the model of strict isolation. We also pairwise compared all models of divergence against the model of panmixia. We again computed type I and type II errors

using 1,000 PODS taken randomly from the prior distribution and running again the same ABC model selection procedure as above.

## Random forest model selection and cross-validations

In parallel to our ABC based model selection and cross-validation procedure we explored the ability of a random forest algorithm (*Breiman, 2001*) to discriminate the different models and to estimate which summary statistics were the most informative. Random forest (RF) is a machine-learning algorithm whose use has recently been advocated for model choice in ABC inference to circumvent curse of dimensionality problems and those linked to the choice of summary statistics (*Pudlo et al., 2014*). This approach is a non-parametric classification algorithm that uses bootstrapped decision trees to perform classification using a set (*p*) of defined predictor variables (here the summary statistics). Multiple (i.e., hundreds to thousands) decision trees are grown and merged together and the ensemble makes up the forest (*Breiman, 2001*). Simulations that are not used in tree building at each bootstrap (the so called out-of-bag simulation OOB) are then used to compute the OOB error rate, which provides a direct method for cross-validation (*Breiman, 2001*; *Cutler et al., 2007*). This method allows reducing the dimensionality of the data (*Cutler et al., 2007*) but also estimating the relative importance of variables (here the summary statistics) through rankings. Variable importance is measured by random permutations of the specified variable in OOB observations and new predictions are then obtained and compared to the original OOB data (*Cutler et al., 2007*). One particularly attracting feature of random forest is its insensitivity to strong correlations and high noise within data (*Pudlo et al., 2014*). In addition, the RF analysis has the advantage of being computationally far less extensive than the ABC cross validation procedure.

We first constructed 6 random forests (one by river) using the randomForestSRC package in R (*Ishwaran et al., 2008*; *Ishwaran & Kogalur, 2015*) allowing for parallelization and fast computations. We grew 1,000 trees on subsets of 50,000 simulated datasets (5%) that were used as a training set. Prior analysis using different numbers of trees and training set sizes indicated that the OOB errors reached stationarity using between 500–1,000 trees (see also Fig. 3), so we did not grow bigger forests that would have required extensive computations. All summary statistics were included to get an estimation of the importance of each variable. This allowed us to estimate the OOB error rate for each comparison, which is similar to a prior error rate in ABC inference (*Pudlo et al., 2014*). Ultimately our forest was used as a prediction tool to compute the probability that our observed data belongs to one of the 5 alternatives models.

## Parameter estimation and cross-validation

Parameter estimation was performed for the best models using nonlinear regressions. We first used a logit transformation of the parameters on the 2,000 best replicate simulations providing the smallest Euclidian distance $\delta$ (*Csilléry, François & Blum, 2012*). We then jointly estimated parameters' posterior probability using the neural network procedure implemented in the abc package. We obtained the best model by weighted nonlinear regressions of the parameters on the summary statistics using 50 feed-forward neural

networks and 15 hidden layers. We performed posterior predictive checks for cross-validation in an attempt to check the ability of our parameter estimates to generate data summary statistics close to the observed summary statistics. For each best model, we selected 10,000 samples drawn from the posterior distribution obtained after parameter estimation (from the abc package) and simulated 10,000 new datasets by using again ms and custom R scripts. We then again plotted the distance between our observed original values and our new simulations and computed the p-value for each statistic as the proportion of values that were larger or smaller than the observed value.

# RESULTS

## Population diversity and divergence

A total of 6 populations pairs (727 individuals in total) were analysed using 13 microsatellites markers. As already observed (*Rougemont et al., 2015*), the averaged allelic richness of river lamprey populations was significantly higher than that of brook lampreys ($Ar_{Lf} = 3.43$, $Ar_{Lp} = 3.116$, $p = 0.0010$, Table 2). However, there was no significant difference in expected heterozygosity between river lamprey (He $_{Lf} = 0.507$) and brook lamprey (He $_{Lp} = 0.46$). ($p = 0.208$). The global genetic differentiation between river and brook lampreys was $F_{ST} = 0.061$ (99%CI [0.044–0.079]) and pairwise $F_{ST}$ ranged from 0 to 0.192 (Table S1). The differentiation among river lamprey populations was significantly lower ($F_{ST} = 0.002$) than among brook lamprey populations ($F_{ST} = 0.109$) ($p = 0.003$). No river lamprey population differed significantly from the others whereas all brook lamprey populations were significantly differentiated (Table S1). The pairwise differentiation between river lamprey and brook lamprey within each river ranged from 0.028 (Oir and Bethune) to 0.091 in the Bresle River and was significant in all cases but the Bethune River (Table S1).

## Model comparison and misclassification

The classical ABC model-choice and random forest approaches generally yielded similar results as detailed in Table 3. In all population pairs, the model of strict isolation (SI) and of ancient migration followed by a period of strict isolation (AM) were clearly rejected. In two population pairs (Aa and Bresle), the best supported model by both model-choice approaches was the SC model. In the Bethune River, the best supported model by both methods was the IM model. In two cases (Hem and Risle), none of the methods was able to accurately discriminate between the two scenarios (SC and IM model). Finally, in the Oir River the two methods gave incongruent results with the model of panmixia (PAN) being the best supported model under the ABC framework, while the RF failed to distinguish between the IM and SC models.

Cross-validations using the ABC framework indicated that the SI and PAN models were correctly classified in all rivers. The PAN model had very low type II errors across all rivers (Table S2) as well as low type I errors in pairwise model comparisons (Table S3). Type II errors for SI and AM comparisons were high (Table S2 and Table S3) but this was a minor concern here as these models were not supported by our data. Cross-validations comparing the two most probable models (IM and SC) indicated a high risk of selecting the IM model

**Table 3 ABC classification (posterior probability) and random forest (RF) prediction of each model of speciation in each river.**

| | MODEL | | | | | | | | | |
| | SI | | IM | | AM | | SC | | PAN | |
| RIVER | ABC | RF | ABC | RF | ABC | RF | ABC | RF | ABC | RF |
|---|---|---|---|---|---|---|---|---|---|---|
| AA | 0 | 0 | 0.27 | 0.39 | 0.01 | 0.06 | **0.72** | **0.54** | 0 | 0 |
| BET | 0 | 0 | **0.45** | **0.57** | 0 | 0.02 | **0.46** | 0.35 | 0.08 | 0.06 |
| BRE | 0.01 | 0.02 | 0.12 | 0.24 | 0.14 | 0.12 | **0.73** | **0.62** | 0 | 0 |
| HEM | 0 | 0 | 0.42 | **0.53** | 0.01 | 0.05 | **0.57** | 0.42 | 0 | 0 |
| RIS | 0 | 0 | 0.46 | **0.47** | 0 | 0 | **0.54** | **0.52** | 0 | 0 |
| OIR | 0 | 0 | 0.15 | **0.46** | 0 | 0.02 | 0.14 | **0.47** | **0.71** | 0.05 |
| Average | 0.00 | 0.00 | 0.31 | 0.44 | 0.03 | 0.05 | 0.53 | 0.49 | 0.13 | 0.02 |

**Table 4 Random forests out-of-bag confusion matrix and classification error.** Data based on 6 random forests, each composed of 1,000 trees based on a trained set of 50,000 simulated predictor variables (summary statistics). The response variable is the demographic model. Proportions of correctly classified demographic models are in bold. The grey italic values represent models with high error rates. Simulation between rivers differed only by the number of individual loci simulated and produced very similar values that were subsequently averaged over each demographic model.

| | Predicted model (Averaged over each river) | | | | | Averaged OOB error rate |
| Observed | AM | I | IM | PAN | SC | |
|---|---|---|---|---|---|---|
| AM | **78.0%** | 16.7% | 2.4% | 0.0% | 2.9% | 21.99% |
| I | 25.2% | **73.6%** | 0.6% | 0.0% | 0.6% | 26.38% |
| IM | 1.6% | 0.2% | **57.2%** | 0.8% | *40.1%* | *42.76%* |
| PAN | 0.0% | 0.0% | 0.2% | **99.7%** | 0.1% | 0.30% |
| SC | 2.1% | 0.3% | *43.6%* | 0.6% | **53.3%** | *47.12%* |

instead of the SC model in all rivers (averaged type II error = 0.4025). In contrast, the risk of selecting the SC model instead of the IM was low (averaged type II error = 0.057).

To gain further insights into the performance of our model selection procedure we tested if a random forest approach could help confirming the validity of the empirically rejected models and distinguishing between the IM and SC models. The RF results confirmed that the models of strict isolation, ancient migration and panmixia were classified with a high accuracy (Table 4, Fig. 3 and Fig. S1). The overall error rate (28.79%) hid very different accuracies depending on models. Average OOB errors in pairwise analyses of IM *versus* SC models were as high as 45% demonstrating that it was generally not possible to correctly classify simulated data in their correct categories (see details in Table 4 and Fig. S1). The estimation of variable importance (Fig. 3 and Fig. S1) indicated that the most informative variables were systematically the mean and variance of $G_{ST}$ and of $\delta\mu^2$, generally followed by the estimators of allelic richness and expected heterozygosity in each population and globally (Fig. 3, Fig. S1 and Table S4).

**Table 5** **Estimates of demographic parameters under the model of ongoing migration (IM) and secondary contact (SC) in each river.**

| River | Model | N$e$ $_{Lf}$ | N$e$ $_{Lp}$ | N$e$ ancestral population | Migration from $Lp$ to $Lf$ | Migration from $Lf$ to $Lp$ | Split time | Time secondary contact |
|---|---|---|---|---|---|---|---|---|
| | | median [95HPD] | median [95HPD] | median [95HPD] | median [95HPD] | median [95HPD] | median [95HPD] | median [95HPD] |
| **AA** | IM | 1310[930–2020] | 410[290–680] | 2290[2260–2350] | 0.0025[0.0018–0.0032] | 0.003[0.0027–0.003] | 268000[246000–282000] | |
| | SC | 1480[760–2550] | 390[270–620] | 1850[1600–2110] | 0.0022[0.002–0.0025] | 0.0032[0.003–0.0033] | 191200[168000–230000] | 89200[61400–108000] |
| **BET** | IM | 1620[1260–2350] | 940[430–1540] | 1650[1500–1780] | 0.0029[0.0027–0.0032] | 0.0035[0.0033–0.0037] | 29800[24200–34200] | |
| | SC | 1930[1280–2630] | 940[570–1420] | 1160[1030–1300] | 0.0024[0.002–0.0027] | 0.0033[0.0032–0.0038] | 322000[265600–396400] | 164600[116000–212800] |
| **BRE** | IM | 1020[440–2450] | 310[120–790] | 1360[440–2370] | 0.0004[0.0004–0.0015] | 0.002[0.0006–0.0039] | 274000[99800–453800] | |
| | SC | 1610[910–2720] | 740[220–1500] | 1440[320–2620] | 0.0017[0.0004–0.0041] | 0.0025[0.0007–0.0046] | 268000[140800–446400] | 20400[3800–124600] |
| **HEM** | IM | 1000[710–194] | 190[130–310] | 2600[2480–2700] | 0.002[0.0017–0.0027] | 0.0042[0.0039–0.0044] | 240000[191200–270000] | |
| | SC | 860[660–1280] | 350[80–880] | 1680[1200–1880] | 0.0031[0.0028–0.0034] | 0.0024[0.002–0.0029] | 278000[231400–328800] | 99400[85400–110000] |
| **RIS** | IM | 840[600–1540] | 620[620–880] | 840[770–930] | 0.0021[0.0015–0.003] | 0.0025[0.0018–0.0031] | 197000[181800–209400] | |
| | SC | 1360[840–2320] | 640[640–1030] | 1010[660–1520] | 0.0036[0.0034–0.004] | 0.0037[0.0036–0.0038] | 226000[168600–320400] | 91200[42600–151800] |
| Average | IM | 1158[788–1710] | 494[318–840] | 1748[1490–2026] | 0.0020[0.0016–0.0027] | 0.003[0.0025–0.0036] | 201760[148600–249880] | [–] |
| | SC | 1448[890–2300] | 612[356–1090] | 1428[962–1886] | 0.0026[0.0021–0.0033] | 0.003[0.0025–0.0037] | 257040[194880–344400] | 92960[61840–141440] |
| **OIR** | PAN[a] | 2050[1940–2180] | | | | | | |

**Notes.**

N$e$, effective population size; $Lf$, *Lampetra fluviatilis*; $Lp$, *Lampetra planeri*.

[a]The *PAN* model is controlled by a single parameter the effective population size of the single population (made of both $Lf$ and $Lp$ backgrounds).

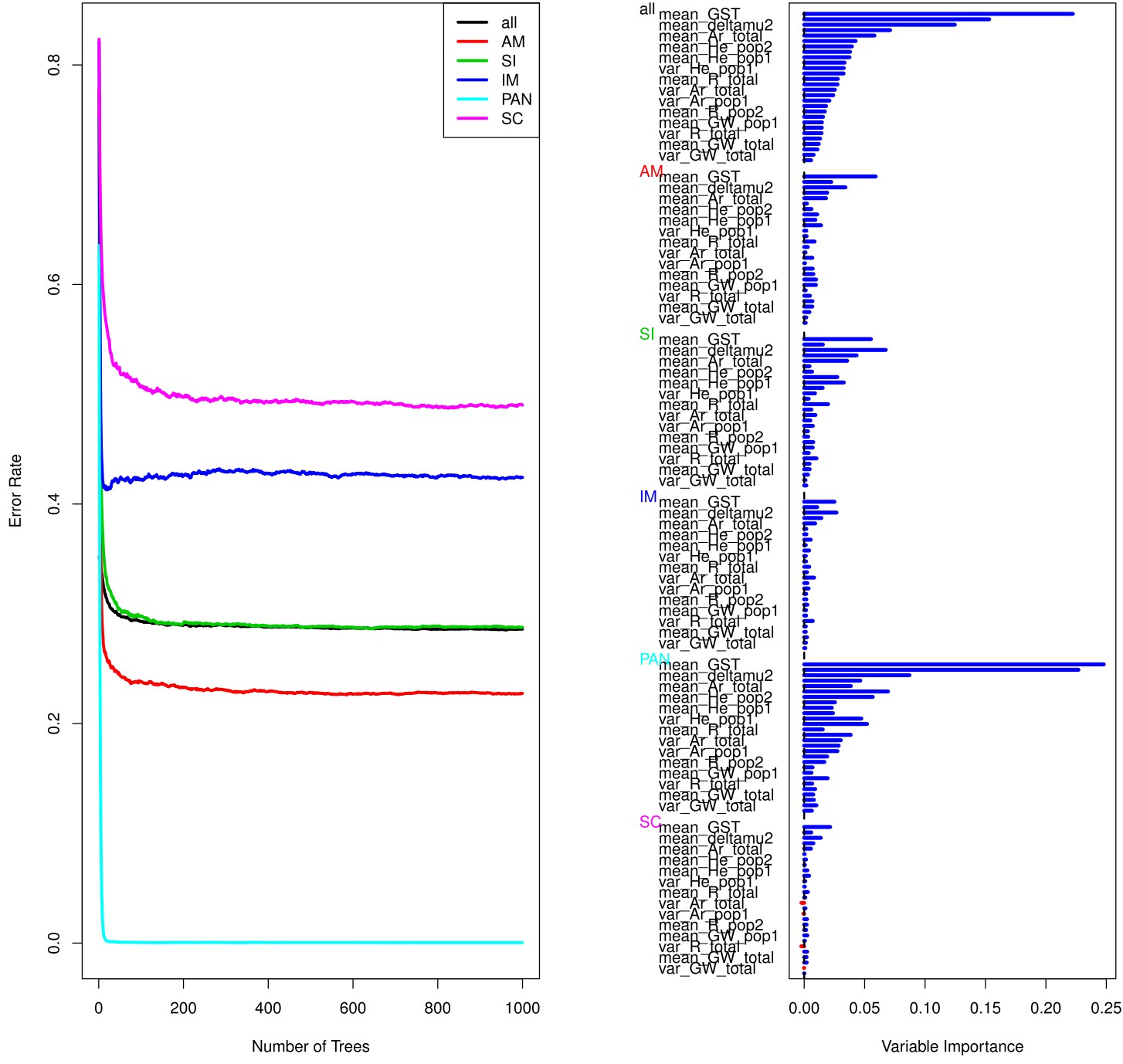

**Figure 3** **Curves of out-of-bag errors rates and estimation of variable importance.** Data based on one random forest, each composed of 1,000 trees obtained from a trained set of 50,000 simulated predictor variables (summary statistics). The response variable is the demographic model. Example taken from the Aa river. Estimation for the remaining rivers yielded similar results and are presented in Table S2 and Fig. S1.

### Parameters estimation from the best models

We estimated the parameters in each population pair for both the IM and SC models that we failed to distinguish and for the PAN model in the Oir population pair (Table 5, see also Table S6 for untransformed values). The accuracy of posterior parameters estimation varied among population pairs, with the Aa, Hem, Bethune and Risle presenting accurate parameters estimation under both the IM and SC model. In general N$e$ estimates from the IM model were slightly more accurate than those from the SC model. Under both models, we generally observed a reduction of N$e$ in both river and brook lampreys as compared to their ancestral populations. Under the IM model, the respective median effective population size (N$e_{Lf}$ and N$e_{Lp}$) of the river lamprey and brook lamprey were on average 1.54 and 5.35 times smaller than the ancestral population. Under the SC model, the (averaged) median effective population sizes were 1.09 and 2.86 times smaller in river lamprey and brook lamprey respectively than their ancestral populations (Table 5). Estimates of N$e$ were on average 2.96 times larger in river lampreys than in brook lampreys under the IM model. Under the SC model estimates of N$e$ were 2.54 times larger in river lampreys than in brook lampreys. Under both the IM and SC models we also noted a slight tendency towards higher migration from river lamprey to brook lamprey with a small deviation from the 1:1 ratio: averaged $m$ ratio = 2.14 and 1.22 under IM and SC respectively but 95% CI's overlapped in 3 out of 5 cases (Table 5). Estimates of divergence time and timing of secondary contact (SC only) yielded variable results and were not always accurately estimated (Table 5). Estimates from the Aa and Hem population pairs were the most accurate under both scenarios. Overall estimates of split times were congruent under the SC model but revealed that populations would have come into secondary contact for a long period with the averaged time of secondary contact representing more than one third of the time since divergence. Finally, simulations of the Oir population pair were summarized by a single parameter (the effective mutation rate of a panmictic population) that was estimated with high accuracy (Table 5).

### Posterior predictive checks

We performed posterior predictive checks in order to assess the ability of the models to accurately reproduce summary statistics close to our observed statistics based on 10,000 simulated datasets drawn from the posterior distribution and computed the robustness of our inference. Under both the IM and SC models we consistently found some statistics that differed significantly from our observed data, indicating that the best proposed models could not reproduce exactly the observed datasets. Genomic variances in heterozygosity and in allelic number are respectively underestimated and overestimated by the best estimated models. In some cases the mean Garza-Williamson index was not accurately reproduced and underestimated (Table S5). Similarly, in three cases, the variance in $F_{ST}$ did not yield accurate results (Table S5 and Fig. S2). Under the PAN model (Oir population pair) we found that the variance in allelic richness and the Garza Williamson index were not accurately reproduced by our data (Table S5). These departures between simulations and observations probably reflect complex features in the real genomic divergence history of lampreys that our models did not capture. For instance, the proposed models do not

allow temporal variation in migration rates or partial linkage to molecular targets of natural selection with various effects, from positive to balancing selection.

## DISCUSSION

Our goal was to test whether we could discriminate alternative scenarios of divergence between river and brook lampreys using a set of microsatellite markers and an ABC approach. We were able to reject the models of strict isolation and of ancient migration. For one population pair (Oir), the model of panmixia received the best support, whereas in other population pairs it was not possible to discriminate divergence with ongoing gene flow from a model of allopatric divergence followed by secondary contact.

### Difficulty in distinguishing between ongoing migration and secondary contact

In spite of the availability of large amount of genetic data and computer resources, few studies have explicitly tested alternative models of divergence (e.g., *Ross-Ibarra, Tenaillon & Gaut, 2009*; *Duvaux et al., 2011*; *Roux et al., 2013*; *Roux et al., 2014*; *Butlin et al., 2014*). While populations may diverge (and eventually become reproductively isolated) under various demographic scenarios, our results indicate that distinguishing between primary differentiation (divergence with gene-flow) *versus* allopatric divergence followed by secondary contact remains difficult when using a limited number of neutral markers, even with advanced computational tools. Indeed, ABC as well as RF cross-validation clearly showed that the two models were wrongly classified almost half the time. The SC model tended to display a larger proportion of simulations wrongly classified into the IM model, a result that can be explained by the higher complexity of this model that displays one supplementary parameter and is inherently more difficult to infer. In contrast, even though the OOB error rate was high in the IM model, it tended to display fewer simulations wrongly classified into the SC model. Given the inherent difficulty to correctly classify the SC model even when it is true, our support for this model in some cases seems conservative and may suggest that it could be the true model under which lampreys have diverged.

Our inability to distinguish between a scenario of isolation with migration and secondary contact is in accordance with theoretical expectations from *Bierne, Gagnaire & David (2013)*. Using a simple modelling approach they showed how genetic environmental associations at neutral markers such as microsatellites can be quickly lost after secondary contacts and then reach migration/drift equilibrium together with a pattern of isolation by distance, which is the pattern observed in the studied lamprey populations (see *Rougemont et al., 2015*). This was indeed expected under the relatively high mutation rate of microsatellite loci that allows neutral equilibrium to be rapidly attained. *Bierne, Gagnaire & David (2013)* applied their model to the well-studied freshwater/marine stickleback system (e.g., *Colosimo et al., 2005*; *Hohenlohe et al., 2010*; *Hohenlohe et al., 2012*), which shares several characteristics with the lamprey system: a single nearly panmictic marine population and small and potentially independent freshwater populations. The application to the stickleback model showed that introgression proceeded independently between the different streams and was strongly asymmetric from the migratory to the resident

populations, which is similar to the pattern we observed here (Table 5) under both the isolation with migration model and the secondary contact model. Although we cannot accurately infer the symmetry of migration, the migration appeared slightly higher from river lamprey to brook lamprey in some rivers (Table 5).

The difficulty in distinguishing the two models is further explained by the fact that when allopatric divergence is short and secondary contacts are very long, the SC model converges to the same signal as that obtained under the IM model. The same difficulty is expected when comparing a model of ancient migration in which migration lasts a short amount of time and is followed by a long period of divergence without gene flow to a model of strict isolation. Accordingly, these models could not be clearly distinguished in our cross-validations (e.g., Table 4).

The failure to reject panmixia in the Oir River can also be explained in the light of *Bierne, Gagnaire & David (2013)* conclusions. It could be linked to the low genetic divergence observed ($F_{ST} = 0.028$) especially given the small number of markers we used, but this pattern of nearly panmixia can also be attributed to a stronger introgression in this system than in all other investigated streams. In this river the mean size of river lampreys (225 mm, $n = 134$) was much smaller than the size observed in other sites (mean = 303 mm, $n = 389$). Assuming that size difference is the most important cause of reproductive isolation (*Beamish & Neville, 1992*) a smaller size difference may facilitate mating of the two ecotypes and subsequent genome swamping. In both cases, inferences from this system based on neutral markers are necessarily difficult as this pattern may be explained by strong gene flow after an isolation period as well as by a very early stage of ongoing divergence.

## Demographic parameter estimations and new insights on lamprey history

We observed asymmetries in effective populations size between river lamprey (averaged median = 1,448; 95% CI [890–2,300] under SC, 1,158 and 95% CI [788–1,710] under IM) and brook lamprey (averaged median = 612; 95% CI [356–1,090]; and 494 95% CI [318–840] under SC and IM respectively). The greater estimates of effective population size in river lamprey reflect the idea that homing is moderate in this species and population size as well as gene flow are large (*Rougemont et al., 2015*; *Spice et al., 2012*) resulting in a situation similar to that observed in the panmictic populations of marine sticklebacks (*Hohenlohe et al., 2010*; *Hohenlohe et al., 2012*). We also observed reductions in effective population size of each resident brook lamprey population as compared to the ancestral populations (averaged median = 1,428; 95% CI [962–1,886]; and median = 1,748 with 95% CI [1,490–2,026] under SC and IM respectively). Such a reduction is expected following the independent river colonization by founding individuals from the resident ecotype (*Bierne, Gagnaire & David, 2013*).

Estimates of divergence times suggested that the two ecotypes may have split around 201,000 years ago (95% CI [148,000–250,000]) under the IM model and around 257,000 years ago (95% CI [195,000–344,000]) under the SC model (assuming a generation time of 5 years, *Potter & Potter (1971)*). Such estimates are rather similar to what was observed in *Dicentrarchus labrax* (*Tine et al., 2014*). Consequently, it seems unlikely

that the divergence was initiated rapidly, following the recent glacial retreats around 10,000–15,000 years ago (*Bernatchez & Wilson, 1998*; *Aldenhoven et al., 2010*) as often proposed to explained ecotypic divergence in various aquatic species (e.g., *Schluter & Nagel, 1995*; *Espanhol, Almeida & Alves, 2007*; *Bracken et al., 2015*; *Mateus et al., 2016*). Importantly, under the SC model, the secondary contact would have started around 92,000 years ago, representing more than one third of the total divergence time between the two species. Such an ancient secondary contact implies that the genetic signature of historical geographic isolation carried by neutral markers may have been lost. In these conditions, neutral markers can converge to the same state than the one observed under primary differentiation explaining again the difficulty of discriminating the two models (*Barton & Hewitt, 1985*; *Charlesworth, Nordborg & Charlesworth, 1997*; *Bierne, Gagnaire & David, 2013*). The SC model implies the accumulation of some Dobzhansky-Muller incompatibilities in allopatry when the two ecotypes started to diverge. While both theory (*Orr, 1995*) and empirical evidence (*Moyle & Nakazato, 2009*; *Matute et al., 2010*; *Wang, White & Payseur, 2015*) predict that DMI should accumulate faster than linearly in time, our result suggest a limited amount of isolation. The period of isolation was certainly too short to allow for sufficient DMI to accumulate and to develop strong barriers to gene flow. This would likely explain the low differentiation observed for mtDNA (*Espanhol, Almeida & Alves, 2007*; *Blank, Jürss & Bastrop, 2008*; *Bracken et al., 2015*) and is fully compatible with the observation of viable F1 hybrids (*Hume et al., 2013*; *Rougemont et al., 2015*).

Another question we started to address is the origin of parallel divergence observed between river lamprey and brook lamprey. The process of recent and independent postglacial divergence with gene flow (parallel divergence) either from standing genetic variation or *de novo* mutations seems a conceptually simple explanation often used to explain divergence between freshwater resident and anadromous (or marine) stickleback (*Schluter & Nagel, 1995*). Similar scenarios have been proposed in lampreys (e.g., *Espanhol, Almeida & Alves, 2007*; *Bracken et al., 2015*; *Mateus et al., 2016*). However, models of divergence with gene flow did not receive higher support than models of secondary contact and divergence time estimates appeared older than the onset of glacial retreats. Alternative scenarios involve either multiple independent secondary contacts between populations inhabiting different refugia or secondary contact between anadromous parasitic and freshwater resident populations each having diverged in a different ancestral place. Under this scenario a spatial re-assortment of alleles involved in non-parasitism would allow the recolonization of neighboring rivers (i.e., the transporter hypothesis; *Schluter & Conte, 2009*; *Bierne, Gagnaire & David, 2013*; *Welch & Jiggins, 2014*). Considering (i) the relatively small scale of investigation and (ii) the nearly panmictic situation in river lamprey; the hypothesis of a spatial re-assortment of ancestral variation by migration between adjacent rivers appears more parsimonious. In this case, the history of divergence would reflect a re-interpretation of the transport hypothesis under a scenario of secondary differentiation rather than primary differentiation. The major conceptual difference is the existence of the brook lamprey background before the recent colonization of rivers. This latter process may have arisen either through hybrid genotypes colonizing rivers or transport of alleles broken up by recombination and at low frequency in the river lamprey background.

While the global divergence implies secondary contact, it is debatable whether the spatial re-assortment process constitutes divergence with gene flow or secondary contact hence the difficulty in distinguishing between the two models. Disentangling these scenarios of primary *versus* secondary differentiation under the transporter hypothesis is challenging and may be addressed by combining genome wide data with historical modelling among multiple pairs of populations. Our results also suggest that the RF method provides a valuable complement to the standard ABC model comparison (*Robert et al., 2011*; *Pudlo et al., 2014*; *Marin et al., 2014*). The two methods provided similar outcomes in terms of model choice and subsequent cross-validations except in one population pair (Oir River). Our ability to distinguish between SC and IM was low in both cases. In line with *Pudlo et al. (2014)* we find that the RF approach possesses a series of advantages over the ABC approach such as (1) fast model choice procedure with simultaneous cross validation through OOB computations, and (2) considerable reduction of computational time. Estimating variable importance can be particularly interesting when a large set of variables are used without prior knowledge about the pertinence of the summary statistics used. The choice of summary statistics is an important process in ABC methodology (*Csilléry et al., 2010*). RF may provide such an objective tool that may be complementary to conventional ABC model choice and cross validation procedure. Note however, that the neural network method provided in the abc package, performed very well and provided similar results to the RF model.

## CONCLUSION AND PERSPECTIVES

Our study shed new light on the demographic processes that have shaped the current genetic makeup of population pairs of European river and brook lampreys. In particular, we were able to reject a scenario of divergence in strict isolation and a scenario of ancient sympatric divergence. The scenario of panmixia was also supported only once and it is thus unlikely to be a generalizable scenario across the species range. However, it was not possible to firmly discriminate the SC or IM models but it is likely that distinguishing between these alternatives scenarios is complicated in cases of ancient secondary contacts, especially when investigations are performed with a limited number of neutral markers. In particular, detecting secondary contact may require sufficiently long allopatric divergence and a time of secondary contact that represents only a small portion of the total divergence time. This study thus illustrates the necessity of explicitly exploring alternatives models of divergence before concluding on the prevalence of rapid parallel speciation (*Bierne, Gagnaire & David, 2013*). In addition, similar analyses may help understanding the divergence between other lamprey species pairs in which interspecific gene flow has been described (e.g., *Ichthyomyzon unicuspis* and *I. fossor*, *Docker, Mandrak & Heath, 2012*). Finally, combining modelling approach with a higher number of markers and allowing for heterogeneous migration rate among loci (e.g., *Roux et al., 2013*; *Roux et al., 2014*; *Tine et al., 2014*), variation of migration rate in time and variation of effective population size along the genome, may allow fine-tuning demographic investigations and provide further insight onto the prevalence of secondary contacts *versus* speciation with continuous gene-flow in nature.

## ACKNOWLEDGEMENTS

We wish to thank M. Navascues for providing great insights onto the joint use of R and ms for microsatellite data. We are also grateful to Jeffrey-Ross Ibarra and two anonymous referees for their insightful comments on the manuscript. We thank the Genotoul bioinformatics platform Toulouse Midi-Pyrenees for providing computing and storage resources.

### Funding

This study was funded by the European Regional Development Fund (transnational programme Interreg IV, Atlantic Aquatic Resource Conservation Project). The funders had no role in study design, data collection and analysis, decision to publish, or preparation of the manuscript.

### Grant Disclosures

The following grant information was disclosed by the authors:
European Regional Development Fund.
Atlantic Aquatic Resource Conservation Project.

### Competing Interests

The authors declare there are no competing interests.

### Author Contributions

- Quentin Rougemont conceived and designed the experiments, performed the experiments, analyzed the data, contributed reagents/materials/analysis tools, wrote the paper, prepared figures and/or tables.
- Camille Roux conceived and designed the experiments, performed the experiments, contributed reagents/materials/analysis tools, reviewed drafts of the paper.
- Samuel Neuenschwander and Guillaume Evanno conceived and designed the experiments, contributed reagents/materials/analysis tools, reviewed drafts of the paper.
- Jérôme Goudet and Sophie Launey conceived and designed the experiments, reviewed drafts of the paper.

### Data Availability

GitHub: https://github.com/QuentinRougemont/MicrosatDemogInference.

### Supplemental Information

Supplemental information for this article can be found online at http://dx.doi.org/10.7717/peerj.1910#supplemental-information.

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
