# Peer review of "Reconstructing the demographic history of divergence between European river and brook lampreys using approximate Bayesian computations"

_PeerJ, doi:10.7717/peerj.1910_

## Round 0.1 · original submission · Minor Revisions

Overall, the paper is a well done ABC analysis and raises some good points about model comparison. Although neither of the reviewers had many criticisms of the paper, I had a number of comments and questions that the authors should review.

See the attached PDF (with the same text, pasted below)

The authors state:

>We focused on pairs highly connected by gene flow previously identified in Rougemont

Am I misundertstanding this statement? It seems to suggest populations were ascertained by previously estimated levels of gene flow. If so, wouldn't this bias the results of the ABC analysis?

Line 157-158: Not all of these are standard stats, please briefly describe the nonstandard ones.

Line 159,199: please make all R code available online (github, figshare, etc.)

Line 181: "scaled"

Line 282: What is the differentiation referred to here? Fst is given on line 277. This is something different?

Am I correct that the variable importance done using Random Forest was estimated using simulated data?

The results were confusing at times. I think I would present the model seleciton, and only discuss errors when relevant to the models selected (i.e. do we care about AM vs SI when neither model was selected?). Error rates for models not selected can be reported in a table but probably aren't important to highlight in the text.

Discussion: do you need to report Ne? It seems that inference based on relative values from the ABC may be valid, but if the assumed value of 1,000 for the ancestral population isn't regarded as very accurate, why use it to estimate Ne for other populations?

Regarding estimation of variable importanince in RF: my (limited) understanding is that this is difficult to do well if variables are highly correlated. Is that not a problem here?

Migration was estimated to be highly assymetric, but some of that is due to differences in Ne. Factoring out Ne, differences in m would appear to be smaller, and the CI for these estimates is high. Given this, how strong is the claims of asymmetrical migration?

I always like to be able to investigate differences between the posterior for parameters and their prior. Would it be possible to include these graphs as a supplement?

Figure 1 should be zoomed-in on the actual populations. It might be helpful to include on this figure Fst between populations, as I found it hard to keep track of while reading.

I was hoping for some explanation of differences among populations. The authors offer details for Oir, but why is Bresle so different from the others?

Similarly, some discussion of the divergence that the authors think is occuring is worthwhile. By analyzing these separately are you assuming 6 independent divergence events? And if so, should the 6 brook lampreys be considered the same thing? Or is it that these 6 represent 1 divergence event and then 6 colonizations of different streams?

A number of the posterior predictive sims suggest poor fits to the summary statistics. Is this not a concern?

·

Basic reporting

The authors have uploaded their analysis scripts as supplements to this manuscript. However, in a few places in the manuscript they refer to their analysis as "available from the authors". This need to be changed to reflect that their scripts are available in supplemental material.

I confess that I am not familiar with lamprey biology. I find the ecotypes very interesting and would like a small amount of additional information. For instance what do the non-parasitic forms feed on? Also the authors mention that most lamprey species are found in pairs, and I would like to see a discussion of whether the conclusions about the history of these two pairs can be applied to other lamprey species pairs.

Experimental design

No issues to report.

Validity of the findings

No issues to report.

Reviewer 2 ·

Basic reporting

The paper reported the pattern of genetic differentiation between two lamprey species: river and brook lampreys. The authors genotyped 13 microsatellite loci for 6 pairs of river and brook lamprey populations. The authors tried to infer the demography incorporating population divergence and migration.

Experimental design

No Comments

Validity of the findings

My biggest concern is that the likelihood values are always high in a model of secondary contact after past isolation (SC model) in the 6 pairs. The 6 sampling locations in Figure 1 seem to be isolated, but almost all the 6 population pairs showed that SC model is highly likely. In the first place, I wondered whether there are some artifacts in their statistical methods but not parallel secondary contacts have really occurred. However, Fst values among river lamprey populations are extremely low (Table S1), Ne in river and migration rate from river to brook populations tend to be high (Table 5). So, I imagined that the 6 river populations would be a huge single population (via gene flow), and after the isolation period between the river and brook populations (both are single populations at that time), the brook populations diverged into the 6 population, then they met the secondary contact. In other words, the 6 brook populations would share the isolation period. I do not have other ideas that explain parallel secondary contacts without the bias of methods or artifacts. It should be worth discussing.

---

## Round 0.2 · accepted · Accept

Please give the ms a final check for grammar etc, as I noticed a few minor problems still. Thank you for the careful attention to reviewer concerns!